# Kinetic Extraction of Fucoxanthin from *Undaria pinnatifida* Using Ethanol as a Solvent

**DOI:** 10.3390/md21070414

**Published:** 2023-07-21

**Authors:** Catarina Lourenço-Lopes, Aurora Silva, Paula Garcia-Oliveira, Anton Soria-Lopez, Javier Echave, Clara Grosso, Lucia Cassani, Maria Fatima Barroso, Jesus Simal-Gandara, Maria Fraga-Corral, Miguel A. Prieto

**Affiliations:** 1Nutrition and Bromatology Group, Analytical and Food Chemistry Department, Faculty of Food Science and Technology, Ourense Campus, Universidade de Vigo, E-32004 Ourense, Spainluciavictoria.cassani@uvigo.es (L.C.); jsimal@uvigo.es (J.S.-G.); 2REQUIMTE/LAQV, Instituto Superior de Engenharia do Porto, Instituto Politécnico do Porto, Rua Dr António Bernardino de Almeida 431, 4200-072 Porto, Portugal; claragrosso@graq.isep.ipp.pt (C.G.); mfb@isep.ipp.pt (M.F.B.)

**Keywords:** fucoxanthin, extraction, *Undaria pinnatifida*, kinetics, biological properties

## Abstract

Fucoxanthin (Fx) has been proven to exert numerous biological properties, which makes it an interesting molecule with diverse industrial applications. In this study, the kinetic behavior of Fx was studied to optimize three variables: time (*t*—3 min to 7 days), temperature (*T*—5 to 85 °C), and concentration of ethanol in water (*S*—50 to 100%, *v*/*v*), in order to obtain the best Fx yield from *Undaria pinnatifida* using conventional heat extraction. The Fx content (*Y*_1_) was found through HPLC-DAD and expressed in µg Fx/g of algae sample dry weight (AS dw). Furthermore, extraction yield (*Y*_2_) was also found through dry weight analysis and was expressed in mg extract (E)/g AS dw. The purity of the extracts (*Y*_3_) was found and expressed in mg Fx/g E dw. The optimal conditions selected for *Y*_1_ were *T* = 45 °C, *S* = 70%, and *t* = 66 min, obtaining ~5.24 mg Fx/g AS; for *Y*_2_ were *T* = 65 °C, *S* = 60%, and *t* = ~10 min, obtaining ~450 mg E/g AS; and for *Y*_3_ were *T* = 45 °C, *S* = 70%, and *t* = 45 min, obtaining ~12.3 mg Fx/g E. In addition, for the selected optimums, a full screening of pigments was performed by HPLC-DAD, while phenolics and flavonoids were quantified by spectrophotometric techniques and several biological properties were evaluated (namely, antioxidant, antimicrobial, and cholinesterase inhibitory activity). These results could be of interest for future applications in the food, cosmetic, or pharmaceutical industries, as they show the Fx kinetic behavior and could help reduce costs associated with energy and solvent consumption while maximizing the extraction yields.

## 1. Introduction

Macroalgae have been used for human consumption for centuries, mostly in Asia. However, in recent decades, their consumption has increased in Western countries thanks to globalization [1]. Macroalgae are a nutritious food, due to their content in essential amino acids and peptides, polysaccharides, polyunsaturated fatty acids, fibers, phenolic compounds, and pigments [2]. *Undaria pinnatifida* (Harvey) Suringar is a brown edible macroalga (Phaeophyceae class), commonly known as wakame. This species is originally from Japan, but it has spread to all oceans, being considered an invasive species in many ecosystems [3]. Despite this last consideration, it has a minimal impact in the ecosystem compared to other species, and currently in many countries is destined for the food industry for the preparation of different dishes. However, in recent years, numerous studies suggest that it is a species rich in fucoxanthin (Fx) [4]. This compound is a carotenoid pigment present in algae that can be found in the chloroplasts, mostly of brown algae, and is responsible for primary light harvesting and transferring energy to the chlorophyll–protein complex [5]. Carotenoids are divided into two groups: carotenes, which are molecules constituted by a 40-carbon skeleton but do not hold oxygen in their chemical structure, and xanthophylls which also have a 40-carbon skeleton but hold oxygen in their chemical structure. Fx belongs to the latter, being an orange xanthophyll, whose main structure includes an allen bond, a conjugated carbonyl in the polyene chain, a 5,6-monoepoxide, and hydroxylated and carboxylated residues [6]. This unique chemical structure is associated with many Fx bioactivities, such as antioxidant, anti-inflammatory, antitumoral, neuroprotective, anti-obesity, and skin protective effects. These properties make Fx a possible candidate for the development of diverse industrial applications as a new bioactive ingredient [7], and could also favor the valorization of the species *U. pinnatifida*.

Nowadays, the commercialization of Fx is still scarce. Even though it can be chemically synthesized, this process is inefficient and complex, especially when compared to its extraction from marine organisms. However, Fx extraction has not yet been standardized, which is necessary to design a practical way to profit from its properties. Regarding extraction for macroalgae, green techniques have many advantages compared to conventional ones. However, conventional heat extraction, also known as maceration, is one of the most used extraction methods because it allows the extraction of an extensive range of molecules by changing and adapting the protocols and variables such as solvent, temperature, time, and agitation [8,9]. In addition, it has an easy scale-up that can be used for several applications in the industry. Thus, even though it requires substantial amounts of solvent and high temperatures that can be translated into vast amounts of energy spent, it is still of interest to optimize this methodology to obtain higher yields while reducing costs and environmental impacts. Regarding the selection of textractive solvents, some of the most recent approaches include the use of bio-solvents, ionic liquids, or deep eutectic solvents as greener and less toxic options. Among some of the natural deep eutectic solvents used for Fx recoveries can be underline mixtures of thymol with octa-, deca-, or dodecanoic acids or lactic acid mixed with choline chloride or aqueous glucose [10,11]. However, they have been less explored than conventional solvents due to their straightforward application with simpler equipment and low initial cost requirements [12,13,14]. Among the conventional organic solvents, those that are polar (acetone, ethanol, or ethyl acetate) are much preferred to recover Fx due to its polar nature. Ethanol has been demonstrated to provide efficient results, especially when used at high temperatures [13,14]. The extraction of Fx through conventional heat extraction has been reported by several studies [15,16,17], but it lacks an optimization of the parameters to increase the extraction yields. Therefore, the aim of this work is to study the kinetic behavior of the Fx molecule and optimize three variables, time (*t*), temperature (*T*), and solvent concentration (*S*), to obtain the best Fx yield from *U. pinnatifida* using a heat-assisted extraction. In addition, for the selected optimums, bioactive compounds and several properties (antioxidant, antimicrobial, and cholinesterase inhibitory activities) were evaluated to estimate the biological potential of the extracts, which could be interesting for industrial applications.

## 2. Results and Discussion

### 2.1. Kinetic Optimization Study

For the choice of the extraction solvent, many factors must be taken into consideration, such as their melting and boiling point, and their polarity, density, and viscosity [11]. Ethanol is an excellent choice because it has a large range between its melting and boiling points (−114 to 78 °C), which allows a large range of extraction temperatures; it has a relatively low viscosity (1.2 cP), providing a more favorable mass transfer from the matrix to the solvent due to its capacity to easily penetrate solid samples; and it has a medium polarity (4.4), which is great to extract compounds like Fx that can be dissolved in mid-polar solvent systems [13]. However, water and ethanolic mixtures with more than 60% water were found to be ineffective in solubilizing Fx [14]. In addition, ethanol is considered to have a low environmental impact and low toxicity and is therefore considered a safe and green solvent to be used in the food industry. Hereby, ethanol or ethanolic solutions are great solvents for the extraction of Fx and its future incorporation in food, cosmetic, or pharmaceutical products. Therefore, the present kinetic study was performed using ethanol as an organic solvent to recover Fx.

Results of the experimental work are detailed in Table 1. Briefly, the maximum amount of Fx extracted was reported at 5 °C using 80% ethanol. In these conditions, after 9680 min (~7 days), it was possible to extract 10.06 mg Fx/g AS dw. Although this is the best result, such a long extraction time is not a realistic parameter from an industrial point of view. Similarly, after nearly 2 days of incubation (2640 min) at 25 °C using 60% ethanol, 6.68 mg Fx/g AS dw was obtained. A more reasonable result was observed at 25 °C with 60% ethanol, where 5.74 mg Fx/g AS dw was obtained after an incubation time of 120 min (2.0 h).

In addition, from these data, some results can be extrapolated regarding the stability of Fx. The best outcomes in terms of ethanol solutions were obtained for concentrations between 60 and 90%. In this range of ethanol concentrations, the use of high temperatures (65 and 85 °C) for long periods of time (1200 min) provided lower concentrations (2.8–3.1 mg Fx/g AS dw, on average) than low temperatures (5–25 °C) which provided 2 times higher Fx concentration values (5.9–6.2 mg Fx/g AS dw, on average). On the contrary, for the same range of ethanol concentrations (60–90%), the application of high temperatures (85 °C) for a short time (30 min) was much more efficient (5.3 mg Fx/g AS dw, on average) than the application of low temperatures (5–25 °C), which recovered half or even less Fx concentration (1.8–3.0 mg Fx/g AS dw, on average) (Table 1).

After the first evaluation of the obtained data, a more thorough analysis was performed to find the optimal extraction conditions for the three responses (*Y*_1_, *Y*_2_, and *Y*_3_), based on the kinetic parameters and their statistical analysis. Figure 1 shows the experimental and model-predicted values at different ethanol proportions for *Y*_1_, and Figure 2 represents the experimental data of the amount of E (*Y*_2_) and model-predicted data (lines) as a function of time (*t*, min). Table 2 shows the parametric values *k* and *r* with their 95% confidence intervals for three responses (*Y*_1_, *Y*_2_, and *Y*_3_) at different temperatures (*T*) and ethanol concentrations (*S*), found by the tool Solver. Only statistically significant values were considered to find the optimal conditions. According to this table, there is a high diversity in values for the different types of responses. For *Y*_1_ (µg Fx/g AS), the range of *k* varied between 0.440 and 8.05 mg Fx/g AS, while the kinetic parameter *r* varied between 0.001 and 0.910 min^−1^. The highest values of parameters *k* and *r* were 8.04 mg/g and 0.910 min^−1^, respectively. Regarding *Y*_2_ (mg E/g AS), the range of *k* fluctuated between 50 and 450 mg E/g AS and between 0.030 and 1.120 min^−1^ for the parameter *r*. The highest values of *k* and *r* were 478 mg E/g AS and 1.120 min^−1^, respectively. Lastly, *Y*_3_ is defined as the amount of Fx per gram of extract (mg Fx/g AS) and stands for the purity of the extract obtained (*Y_1_/Y_2_*). The values of parameter *k* varied between 2 and 62 mg Fx/g AS, and the values of *r* oscillated between 0.002 and 0.650 min^−1^. The highest values of *k* and *r* were 61.4 mg Fx/g E and 0.648 min^−1^, respectively.

The different error measures (MAE, RMSE, and RMSE-MAE) for the three responses (*Y*) at different values of *T* and *S* are shown in Appendix A. These parameters can evaluate if the experimental data fit the first-order kinetic Equation (1). Thus, MAE and RMSE evaluate the errors between empirical and experimental values. From Appendix A, the MAE values were below 500 and 150 for RMSE-MAE values at any *T* and *S* for *Y*_1_. According to these results, the magnitude of fitting errors for *Y*_1_ was not significant and in the case of *Y*_2_ and *Y*_3_, the magnitude of errors was generally low. Hence, the observed values are close to the model-predicted values. Therefore, the use of the mathematical Equation (1) for the fitting of experimental data could be a good method, since it presents good reliability.

Additionally, the trend and dependence of the kinetic parameters (*k* and *r*) in relation to *T* and *S* are shown in Table 3. In the case of *Y*_1_, the parameter *k* did not show relevant variations when different *T* were used and *S* remained constant. Hence, *k* is independent of *T*. However, the parameter *r* depends on *T*. In fact, the higher the *T*, the higher extraction rate for any *S*. The highest value of *r* was found when applying 80% of ethanol and 85 °C. On the other hand, when *T* stays constant, *k*_max_ is reached, when applying average ethanol proportions (70%). In fact, when using medium proportions of ethanol (70–80%) at any temperature, the most significate *k* values were found with values above 3.5 mg/g (Table 2). For this reason, *k* depends on *S*. Regarding the other parameter, big variations were not found in *r* when varying *S*. Hence, this kinetic parameter is still constant (does not depend on *S*) for *Y*_1_. Regarding *Y*_2_, the *k* values did not undergo substantial changes at varying *T* when using the same *S*. According to this, *k* is independent of *T*. However, the values of *r* depend on *T*, i.e., the *r* is higher when *T* increases. The maximum *r* values are reached when applying temperatures above 65 °C in which their values were above 0.200 min^−1^ (Table 2). In the case of keeping the same *T*, the parametric value *k* decreases when increasing *S*. Therefore, the most relevant *k* values were found when using low–medium ethanol proportions (<80%) for any temperature in which their values varied between 400 and 480 mg E/g AS (Table 2). However, it is seen that the values of *r* remain unchanged with the variation of solvent, i.e., *r* is independent of *S*.

Finally, in the case of *Y*_3_, *k* is still constant for different temperatures when *S* is unchanged. In addition, when using high temperatures (>65 °C), the values of *r* are higher, highlighting values above 0.300 min^−1^ (Table 2). Therefore, *r* depends on *T*. In the case of *T* being constant, it is seen that both *k* and *r* are independent of *S*, i.e., when varying S, both parameters are still unchanged.

At an industrial scale, the ideal conditions are based on the process in which the maximum Fx concentrations (*k*) are obtained with the highest possible extraction rates (high values of *r*), i.e., the best conditions are those ranges of *T* and *S* in which the values of *k* and *r* are the highest possible. So, according to Table 2 and Table 3, using 45 °C with 70% ethanol and 85 °C with 80% ethanol allowed the extraction of maximum Fx concentrations (*Y*_1_), in which the values of the kinetic parameters were 5.245 mg Fx/g AS and 0.105 min^−1^, and 5.096 mg Fx/g AS and 0.910 min^−1^, respectively. In addition, *R*^2^ for these conditions was quite good (above 0.950). However, for the errors, the value of RMSE-MAE was quite low for *T* = 45 °C and *S* = 70% (31.8). Nonetheless, the value for the condition *T* = 85 °C and *S* = 80% was higher (120.9) (Appendix A). Regarding *Y*_2_, the best conditions to obtain the highest possible yields were when using *T* = 65 °C with 60 and 70% of ethanol. In fact, the values of *k* and *r* were 450 and 437 mg E/g AS, and 0.733 and 1.120 min^−1^, respectively (Table 2). Moreover, *R*^2^ was above 0.999 for both conditions and the RSME-MAE values were 3.0 and 2.2, respectively (Appendix A). Regarding the purity (*Y*_3_), the conditions required to obtain highly pure Fx were 65 °C and 85 °C with 100% of ethanol, in which the values of *k* and *r* were 24 and 29 mg Fx/g AS, and 0.185 and 0.648 min^−1^, respectively (Table 2). Nonetheless, their *R*^2^ values were low (0.744 and 0.691, respectively). For this reason, despite the values of RMSE-MAE of both conditions being low (1.3 and 2.0, respectively), the mentioned conditions are not adequate. Thus, the better conditions for *Y*_3_ are *T* = 45 °C with 70 and 80% of ethanol, in which the parametric values were 12.3 and 11.2 mg Fx/g AS, and 0.155 and 0.171 min^−1^, respectively (Table 2). Moreover, in this case, *R*^2^ was above 0.950 for both conditions and the RMSE-MAE value was below 0.25 (Appendix A).

In addition to the above, high extraction rates (*r*) also show low extraction time (*t*). Through Equation (2), it is possible to calculate *t* at a certain *T* and *S*. Thus, for example, the best conditions to extract the maximum Fx amount (*Y*_1_) are at 45 °C and 70% ethanol and *t* = 66 min or 85 °C and 80% ethanol and *t* ~8 min, in which *Y*_1_ were ~5.24 and ~5.09 mg Fx/g AS, respectively. So, for *Y*_1_, it is possible to choose lower extraction times with elevated temperatures or slightly longer extraction times with average temperatures. Regarding *Y*_2_, the highest yield was obtained at *t* ~10 min for the condition of *T* = 65 °C and *S* = 60%, or *t* = ~6 min when using a temperature of 65 °C and 70% ethanol, in which *Y*_2_ were ~450 and ~437 mg E/g AS, respectively. In this case, both could be the proper conditions since their differences of *t* were insignificant. Finally, for *Y*_3_, the best purity was seen for *t* = 45 min at *T* = 45 °C and *S* = 70%, or *t* = 40 min when using 45 °C and 80% ethanol, in which *Y*_3_ were ~12.3 and ~11.2 mg Fx/g E, respectively. In this case, the *t* difference was also insignificant between these conditions, with both being optimal. However, at an industrial scale, some industries choose extraction processes and conditions where the ethanol % is lower and, therefore, more beneficial in economic and environmental terms, reducing the production of solvent wastes. In this sense, the solid:liquid (S:L) ratio may also be adjusted to maximize the use of the solvent, since the S:L suggested in the present work is hardly scalable in industrial sectors. Furthermore, energy consumption is also considered, giving preferential choice to the use of intermediate temperatures (25 °C or 45 °C) and shorter extraction times. Additionally, the use of remarkably high temperatures (85 °C) could lead to the evaporation of the organic solvent, when the temperature is above its boiling point (T_m_ = 78 °C), resulting in a reduction of the extraction process’s effectiveness and a higher intoxication risk for the employers. And lastly, depending on time availability, a slightly longer *t* can be preferential, since the extracted amount, yield, or purity of Fx still is almost unchanged and it allows the use of lower *T* or *S*. Ultimately, the choice of optimal conditions is dependent on the industry’s interest.

### 2.2. Bioactive Analysis of the Optimal Values

The three optimums of the three responses were assessed for the content of several bioactive compounds (pigments, phenolic, and flavonoid content) and for several biological properties. These results are shown in Table 4.

#### 2.2.1. Bioactive Compounds

The optimal extracts were injected in HPLC-DAD to identify and quantify the pigments present. In total, five peaks were detected, as seen in Figure 3 and Appendix A, identified and quantified using three standards and also by comparison with previous identification studies [18,19]. The first peak detected showed a retention time of 19.1 min and was identified as chlorophyll c [18,20]. The peak was quantified with the chlorophyll a calibration curve, reporting 0.17, 0.02, and 0.04 mg/g dw, for *Y*_1_, *Y*_2_, and *Y*_3_, respectively. The second peak was detected at 22.1 min and was identified as Fx. Regarding the Fx content, for responses *Y*_1_, *Y*_2_, and *Y*_3_, 4.53, 3.82, and 4.42 mg Fx/g dw were quantified, respectively. These results are far superior to the ones previously described in the literature. A heat-assisted extraction from *U. pinnatifida* obtained 0.7 mg/g dw of Fx using ethanol at room temperature for 1 h [15], while another study reported 2.67 mg/g dw using methanol at room temperature for 96 h [16]. At a retention time of 22.9 min, a compound with an absorbance spectrum like Fx was quantified. This pigment is likely a derivative from the Fx molecule and was previously thought to be fucoxanthinol [18,19]. The Fx derivative was quantified using the Fx calibration curve and obtained 0.24, 0.27, and 0.26 mg/g dw, for *Y*_1_, *Y*_2_, and *Y*_3_, respectively. The fourth pigment detected was lutein with a retention time of 23.9 min. This pigment was detected in low amounts and was under the quantification limit for *Y*_2_. For *Y*_1_ and *Y*_3_, 0.02 and 0.01 mg/g dw were quantified, respectively. Lastly, the peak with a retention time of 25.9 min was identified as chlorophyll a, and amounts of 0.14, 0.07, and 0.25 mg/g dw were obtained, for *Y*_1_, *Y*_2_, and *Y*_3_, respectively.

Apart from the pigment identification and quantification, phenolic compounds and flavonoids were also quantified, using spectrophotometric techniques. The TPC analysis showed 3.43, 3.26, and 2.04 mg GAE/g dw for the three responses, respectively. These results are similar to other studies that obtained 3.02 mg GAE/g, in a hot water maceration extract at room temperature for 24 h [21]. Additionally, the present study showed some improvements to previous studies where a standard maceration of *U. pinnatifida* obtained 0.6 mg GAE/g dw [22]. The TFC showed slightly more variation between optimums, with 90.47, 54.17, and 86.20 µg QE/g dw, for *Y*_1_, *Y*_2_, and *Y*_3_, respectively.

#### 2.2.2. Antioxidant Activity

The antioxidant activity of the optimums was evaluated using five different assays: DPPH^•^ scavenging activity, ABTS^•+^ scavenging activity, crocin bleaching assay, ^•^HO scavenging activity, and ^•^NO scavenging activity, and can be seen in Table 4 and Figure 4. In Figure 4A, the dose–response curve is represented for each optimum and for the five in vitro assays aiming to establish the Fx-rich extracts antioxidant profile. The curves obtained showed that the extracts behave in a dose-dependent manner. Additionally, the experimental data fit the Weibull model effectively, allowing for the establishment of the EC_50_ parameter with a 95% confidence level. The coefficient of determination found was always greater than 0.9. The chemical structure of the Fx molecule is considered responsible for its antioxidant activity, due to its allenic bond and acetyl functional group. In fact, numerous studies have confirmed the effectiveness of this molecule against oxidative stress, in both in vitro and in vivo studies [7].

For the DPPH^•^ assay, the results obtained for the EC_50_ parameter were 3.43, 3.36, and 2.04 mg/mL, for *Y*_1_, *Y*_2_, and *Y*_3_, respectively. Specifically, *Y*_3_ obtained better results, which can be associated with its higher purity compared to *Y*_1_ and *Y*_2_. Nevertheless, the three results were slightly higher than the ones obtained for the quercetin control that obtained an EC_50_ value of 0.80 mg/mL. In this case, the extracts were not as effective as the ones reported in previous studies, which reported lower EC_50_ values, but the extracts were obtained in different experimental conditions [23,24,25]. In the ABTS^•+^ assay, the best results were once more obtained for the *Y*_3_ extract, with an EC_50_ value of 0.92 mg/mL. However, comparable results were found for *Y*_1_ and *Y*_2_, with EC_50_ values of 1.00 and 2.67 mg/mL, respectively. In relation to the DPPH results, previous studies on a Fx-rich acetonic extract have reported lower EC_50_ values, such as 0.49 µg/mL [24] and a 33.54 µg/mL EC_50_ value obtained from Fx-purified *Laminaria japonica* methanolic extract [26]. For the Crocin assay, all extracts inhibited the oxidation of the crocin molecule and therefore protected it from discoloration, but the EC_50_ value varied slightly between optimums, being 11.52, 7.01, and 12.13 mg/mL, for *Y*_1_, *Y*_2_, and *Y*_3_, respectively. The result obtained for *Y*_2_ is similar to that of the control quercetin (EC_50_ value of 6.67 mg/mL). For this assay, a previous study conducted with rich Fx extracts obtained from *U. pinnatifida* reported an EC_50_ value of 13.47 µg/mL [24], ~1000 times more effective than the results obtained in the present study, which may be due to differences in the extraction conditions. The scavenging activity of the ^•^HO radical reported EC_50_ values of 2.00, 1.76, and 0.44 mg/mL, for *Y*_1_, *Y*_2_, and *Y*_3_, respectively. In this case, the *Y*_3_ extract obtained results as good as the ascorbic acid (EC_50_ = 0.45 mg/mL), which was used as the control for this assay. In an earlier study, the Fx obtained by methanolic extraction of *U. pinnatifida* followed by purification was highlighted as a hydroxyl scavenger, with an EC_50_ value of 0.14 mg/mL. The authors also postulated that fucoxanthinol, a hydrolysis product of Fx, has a much lower hydroxyl radical scavenging ability, with an EC_50_ value of 1.11 mg/mL [25]. Lastly, the least favorable antioxidant activity results were obtained for the scavenging of the ^•^NO radical, with EC_50_ values of 11.76, 12.65, and 10.48 mg/mL, for *Y*_1_, *Y*_2_, and *Y*_3_, respectively. In this assay, no results were comparable to those obtained for ascorbic acid (EC_50_ = 0.18 mg/mL), even though previous findings imply that Fx may have anti-inflammatory properties and be useful in the management of inflammatory illnesses [27].

All together, these findings demonstrate that the three optimum *Y*_1_, *Y*_2_, and *Y*_3_ possess antioxidant action, which could be attributed to the Fx content, as has been suggested by previous research [28]. Moreover, apart from the crocin assay, the *Y*_3_ was the most active extract across all assays, showing that purity matters in terms of antioxidant ability.

#### 2.2.3. Antimicrobial Activity

In the food sector, where the expansion of shelf life and consumer safety are critical, the ability of substances to interact with the environment and reduce the emergence of dangerous bacteria is essential. In that regard, the antimicrobial capacity of the three optimal extracts was tested by the plate diffusion method against five foodborne bacteria. Lactic acid was selected as the positive control, since the antibacterial effects of organic acids are well known and well researched [29].

DMSO was used as a negative control in all tests and no inhibition zone was detected. Lactic acid at 40% m/V was chosen as the positive control. This choice was made because lactic acid, as a fermentation product, is a natural antimicrobial agent in food, with the extract developed aiming for human consumption, and all bacteria assessed were foodborne pathogens.

The results are presented in Table 4 and Figure 4B. The results show that *Y*_1_, *Y*_2_, and *Y*_3_ extracts can disrupt bacterial growth of Gram-positive bacterium. For *S. aureus*, the inhibition halos reached 9.14 ± 1.05, 11.94 ± 1.70, and 11.40 ± 0.09 mm, respectively (Table 4). Similar inhibition halos were seen for *B. cereus*, being 10.97 ± 1.06, 9.73 ± 0.19, and 11.46 ± 0.40 mm, respectively. Regarding the inhibitory effect on Gram-negative bacteria, the extracts successfully inhibited the growth of *P. aeruginosa* (inhibition halos of 12.84 ± 0.62, 12.69 ± 1.79, and 11.96 ± 0.75, for *Y*_1_, *Y*_2_, and *Y*_3_) and *S. enteritidis* (inhibition halos of 13.70 ± 2.70, 12.76 ± 2.3, and 10.78 ± 1.11, correspondingly), but no inhibition effect was reported against *E. coli*. These results are similar to those obtained previously [24,30,31]. In addition, there is strong evidence that Fx is more effective against Gram-positive bacteria than Gram-negative bacteria, as reported by several studies [32,33]. The slight difference between the performances of the optimums may be related to the fact that the fucoxanthin content is higher in all of them, suggesting that the differences in the extraction conditions (in the tested range) are not crucial for the antimicrobial capacity of the final extracts.

It is important to emphasize that, when compared to the positive control, the obtained extracts were remarkably active against the microbial strains. The inhibition zones of *S. aureus* and *B. cereus* were 67.04 and 56.15% of those observed for the control, respectively (Table 4), despite the extract mass administered, which was 24-fold less than the mass of lactic acid. The same was verified on the inhibition of *P. aeruginosa* (61.20%) and *S. enteritidis* (68.02%). The study conducted by Karpiński and Adamczak, although not directly comparable due to differences in methods, is in agreement with the present work. Specifically, their results revealed an impressive antimicrobial capacity of commercial Fx, reporting inhibition halos ranging from 7.2 mm (*Proteus mirabilis*) to 12.2 mm (*Streptococcus agalactiae*) [31]. Considering this, Fx-rich extracts obtained from *U. pinnafitida* may have considerable potential in the research of new antimicrobials.

#### 2.2.4. Cholinesterase Inhibitory Capacity

Cholinesterase inhibitors are presently the primary pharmaceutical strategy for treating Alzheimer’s disease. Acetylcholine, a neurotransmitter thought to be involved in the pathophysiology of this disease, is blocked primarily by AChE and secondarily by BuChE [34]. The inhibition of these enzymes would increase the amount of acetylcholine in the synaptic clefts, which would lead to the reduction of neurodegeneration, combating Alzheimer’s disease and other neurodegenerative disorders [35,36]. In this sense, Fx has been pointed out as a potential cholinesterase inhibitor that acts through the Nrf2 and Akt signaling pathways [37,38,39]. As can be observed in Table 4 and Figure 4C, the inhibitory effects of the optimums were heterogeneous. The percentages of AChE inhibition were 25.75 and 32.85%, for *Y*_1_ and *Y*_3_, respectively. On the other hand, the *Y*_2_ and *Y*_3_ extracts show residual inhibition of BuChE and no inhibitory effect was seen for the *Y*_1_ optimum. These results were not comparable to those of the control galantamine. Compared with previous studies, the results of AChE inhibition (25.27%) were like those reported in the present study, but higher BuChE inhibition was reported (37.70%) [24]. However, the extracts were obtained in different conditions, using acetone as the extraction solvent. Neuroprotective effects of fucoxanthin have also been reported in in vivo models. For example, Fx effectively protected against scopolamine-induced cognitive impairments in mice, working as the AChE inhibitor (EC_50_ value of 81.2 mM) [40]. Similarly, neuroprotective effects of Fx have been reported in traumatic brain injury, attenuate cerebral ischemic/reperfusion injury, middle cerebral artery occlusion, and oxygen-glucose deprivation and reoxygenation models [38,39]. In addition, oxidative stress is well known to play a significative role in the evolution and physiopathology of neurological disorders. Therefore, the neuroprotective and antioxidant properties of fucoxanthin make this compound a promising candidate for therapeutic investigation in the treatment of neurological diseases.

## 3. Materials and Methods

### 3.1. Samples Collection and Preparation

For this study, a brown alga from the Phaeophyceae family, *U. pinnatifida*, was selected. The alga was manually recollected from Galician coastlines and provided by Algamar (www.algamar.com). Species identification was performed and double-checked by expert biologists both from Algamar and our laboratory, Paula García-Oliveira and María Fraga-Corral. After receiving the fresh alga, it was washed with water, lyophilized (LyoAlfa 10/15 from Telstar, Madrid, Spain, UVigo), pulverized (~20 mesh), and stored in container bags at −80 °C until its use.

### 3.2. Optimization Study for the Extraction Maximization of Fx

#### 3.2.1. Kinetic Extraction with Ethanol

For the extraction of Fx, a kinetic extraction was performed using heat-assisted extraction of *U. pinnatifida* at a solid Tb–liquid ratio of 30 g/L of ethanol. The three independent variables assessed were: time (*t*, 3 min to 7 days), temperature (*T*, 5 to 85 °C), and concentration of ethanol in water as the extraction solvent (*S*, 50 to 100%, *v*/*v*). The initial concentration of ethanol was 96%; however, to facilitate data representation, we labeled it as 100%, and so the following aqueous solutions (50, 60, 70, 80, and 90%) had real ethanol concentrations of 48%, 57.6%, 67.2%, 76.8%, and 86.4%, respectively. The ethanol concentrations used were 50, 60, 70, 80, 90, and 100%, and for each concentration, the following combinations of *T* and *t* were applied: 5 °C (30, 120, 480, 1200, 2640, 4200, 5700, and 9680 min), 25 °C (15, 30, 120, 480, 1200, 1680, and 2640 min), 45 °C (3, 5, 15, 60, 210, 1200, 1680, and 2640 min), 65 °C (3, 5, 15, 30, 60, 120, 210, 480, 1200, and 1680 min), and 85 °C (3, 5, 15, 30, 60, 120, 210, 480, 1200, and 1680 min). As observed in these experimental combinations, higher temperatures were tested at shorter incubation times, while lower temperatures were assessed up to the longest times. With this design, a total of 258 experimental points were generated. Amber bottles used for the extraction were sealed during incubation and cool-down times to prevent evaporation issues. After applying the extraction conditions to the dried alga sample, the extracts were centrifugated at 8400 rpm for 7 min to eliminate any algae residue. The supernatant was filtered with nylon syringe filters with Ø 0.22 μm and divided into two different subsamples for later analysis: (1) Fx determination for HPLC; and (2) Extract dry weight (E) determination. With the combination of this information plus the mathematical model optimization, optimal condition values were determined and finally subjected to further bioactive analysis.

#### 3.2.2. Chemical Analysis through HPLC

For each of the experimental conditions assessed (258 points), the Fx content was analyzed by HPLC-DAD and expressed in mg Fx/g E dw. To quantify Fx, the HPLC equipment used was a Waters HPLC equipped with a Waters 600 controller and pump, Waters 2996 photodiode array detector (1.2 nm optical resolution), Waters 717 plus autosampler, and an AF in-line degasser. The column used for the stationary phase was a Waters Nova-Pak C18 column (150 × 3.9 mm, WAT 088344), stabilized at 25 °C.

The method applied for the determination of Fx was the one developed previously, disclosed in another work [19] and briefly described next. The mobile phase solvents were 5 mM ammonium acetate in milli-Q water (A), 5 mM ammonium acetate in methanol (B), and pure ethyl acetate (C). All solvents were HPLC grade. The flow rate was set to 0.5 mL/min and the injection volume was 50 µL. The established eluent gradient started with 5% A and 95% B up to min 8, 50% B and 50% C up to min 20, 50% A and 50% B up to min 35, and 30% A and 70% B until the end of the run at 40 min.

Fx, lutein, and chlorophyll standards were bought from Sigma. Fx and Fx derivative were quantified using the calibration curve of Fx standard (*y* = 3.10 × 10^8^*x* + 2.73 × 10^5^, *R*^2^ = 0.9944); lutein was quantified with lutein standard (*y* = 1.11 × 10^9^*x* + 2.06 × 10^5^, *R*^2^ = 0.9862); and chlorophylls a and c were both quantified based on the calibration curve for the standard of chlorophyll a (y = 3.35 × 10^8^*x* − 3.86 × 10^4^, *R*^2^ = 0.9993).

#### 3.2.3. Extract Dry Weight Determination

For the determination of the extracts’ dry weight to find the extraction yield, 5 mL of each extract was transferred to previously prepared crucibles. These were left to dry at 104 °C, for 24 h in the dark, until complete dryness. The crucibles were then weighed, avoiding air moisture with the help of a desiccator, and the percentage of dry residue with respect to the original mass, expressed the extraction yield.

#### 3.2.4. Responses Obtained for Analysis

The Fx content (*Y*_1_) was determined through HPLC-DAD and expressed in µg Fx/g of algae sample dry weight (AS dw). Furthermore, through dry weight analysis, the extraction yield (*Y*_2_) was also determined and was expressed in mg extract (E)/g AS dw. The purity of the extracts (*Y*_3_) was determined and expressed in mg Fx/g E dw.

#### 3.2.5. Mathematical Modeling for Optimization Purposes

The models applied for the prediction and optimization of Fx were previously applied in a prior work [24] and are briefly described next. The extraction conditions that maximize the process’s responses can be mathematically predicted by a first-order structure that can be fitted with a two-parameter equation, as next described in Equation (1):(1)$$R=k(1-e^{-rt})$$
where *R* represents one of the responses used in the analysis (*Y*_1_, *Y*_2_, and *Y*_3_) with their respective dimensional units (µg Fx/g AS, E/g AS, and mg Fx/g E dw, respectively); *k* is related to the maximum amount of each of the responses analyzed (*Y*_1_, *Y*_2_, and *Y*_3_) for each *T* and *S* value; and *r* (min^−1^) is the rate constant that provides information about the extraction rate.

To calculate the extraction time (*t*) at a certain *T* and *S* value, the following Equation (2) was used:(2)$$t=\frac{Ln(2^{n})}{r}$$
where *r* is the kinetic constant and *n* is the number of semi-extraction periods elapsed. In this study, 10 semi-extraction periods were used, which means the time needed to extract 99.9% of each of the responses analyzed (*Y*_1_, *Y*_2_, and *Y*_3_).

SigmaPlot 14.0 program was used to obtain the graphical representation of the results. All adjustment procedures were performed using a Microsoft Excel spreadsheet following the next four phases: (1) Determination of the parametric values by using the non-linear method of least squares (quasi-Newton) provided by the macro Solver in Microsoft Excel 2003 [41]; (2) Determination of the confidence intervals of the parameters using ‘*SolverAid*’ [42]; (3) Model consistency by Fisher’s F test (α = 0.05) [43]; and (4) Verification of the uniformity of the model with the macro ‘*SolverStat*’ [44], taking into account the prediction uncertainties described by the correlation coefficient of determination (*R*^2^) value and the adjusted value (*R*^2^*adj*).

### 3.3. Bioactive Analysis of the Optimal Values

#### 3.3.1. Analysis of Secondary Metabolites

##### Total Phenolic Content (TPC)

For the determination of the TPC, 25 µL of the 3 optimal extracts and 7 dilutions of each were added to a microplate with 96 wells. The Folin–Ciocalteu (1:10) reagent was added to all wells (125 µL) and the plate was left to rest for 3 min at room temperature. After, 100 µL of Na_2_CO_3_ (7.5% *m*/*v*) was added to every well [22]. The microplate was then incubated at room temperature for 120 min in the dark. The absorbance was measured at 765 nm using a Synergy™ HTX microplate reader. This assay was made in triplicate and the blank was made with 25 µL of distilled water. The results were expressed in mg GAE/g dw.

##### Total Flavonoid Content (TFC)

The TFC was determined for the three optimal extracts by spectrometric analysis [22]. For that, 100 µL of the sample was added to a Khan tube, and 400 µL of distilled water and 30 µL of NaNO_2_ (5% *m*/*v*) were added. The mixture was left to rest for 5 min. Then, 30 µL of AlCH_3_ (10%) was added and, after 1 min, 200 µL of NaOH (1 M) and 240 µL of distilled water were added. The mixture was vortexed, 250 µL was added to a microplate with 96 wells, and the absorbance was read at 510 nm using a Synergy™ HTX microplate reader. The assay was made in triplicate and the blank was made using 100 µL of solvent instead of the sample. The results were expressed in µg QE/g dw.

#### 3.3.2. Antioxidant Determination

##### DPPH Radical-Scavenging Activity

The antiradical activity was determined through the ability to scavenge DPPH• radical [24]. For that, 50 µL of each optimum and 7 dilutions of each were added to a microplate with 96 wells. Then, 200 µL of the DPPH reagent was added and the plate was left to rest at room temperature in the dark for 1 h. After that, the absorbance was measured at 515 nm using a Synergy™ HTX microplate reader. The assay was made in triplicate and the blank was made with 50 µL of methanol.

##### ABTS^•+^ Radical-Scavenging Activity

For the evaluation of the antiradical activity through the ABTS^•+^ radical assay, a reagent was created by the reaction of 19.3 mg of ABTS^•+^ dissolved in 5 mL of distilled water and 37.8 mg of potassium persulfate (K_2_S_2_O_8_) in 1 mL of distilled water. Of the second mixture, 88 µL was added to the ABTS^•+^ mixture to form the final solution [24]. The solution was left to rest at room temperature and in the dark for 16 h. This solution was then diluted at a 1:10 ratio with ethanol to obtain an absorbance between 1.3 and 1.4 units, measured at 734 nm. In a microplate, 190 µL of the previously prepared ABTS^•+^ radical solution was added to 10 µL of each optimum (diluted in 7 different concentrations with the respective solvent) and left incubating for 6 min at room temperature. The absorbance at 734 nm was measured using a Synergy™ HTX microplate reader. The assay was made in triplicate and the blank was made using 50 µL of ethanol.

##### Crocin Bleaching Assay

The determination of the crocin bleaching assay was performed using a stock solution that had 5 mg of crocin in 25 mL of distilled water at 40 °C, as well as a solution of 75 mg of AAPH in 5 mL of ultrapure water at 40 °C [45]. After the complete dissolution of both parts, they were mixed and with this preparation, an absorbance of ~1.4 was obtained. The solution was used immediately after preparation, where 250 µL was added in each well of the microplate previously prepared with 50 µL of 7 concentrations of the extracts in this study, as well as a water blank. The absorbance was read at min 0 and the microplate was left to incubate, in agitation and protected from light at 37 °C, and then read at 30 min intervals until complete discoloration of the blank, at 450 nm, using a Synergy™ HTX microplate reader. The assay was made in triplicate.

##### Scavenging Activity of the Hydroxyl Radical

The scavenging activity of hydroxyl radical (HO^●^) was performed based on the salicylic acid method, as described previously [46]. Briefly, 70 µL of the optimal extract dissolved in demineralized water, 70 µL salicylic acid 9 mM, and 70 µL of ferrous sulfate 9 mM were added to each microplate well. The reaction started by adding 70 µL of hydrogen peroxide 9 mM. Afterwards, the mixture was incubated at 37 °C for 30 min. For the negative control, the extract was replaced with water and ascorbic acid was used as the positive control. The absorbance was read at 510 nm using a Synergy™ HTX microplate reader. The test was carried out in triplicate.

##### Scavenging Activity of the Nitric Oxide Radical

The ^●^NO scavenging activity was evaluated based on a diazotization reaction, as described in a similar application [47]. The optimal extracts were dissolved in potassium phosphate buffer 0.1 M pH = 7.4 and were incubated with sodium nitroprusside 20 mM for 1 h at room temperature. After that, the Griess reagent (1% sulphanilamide and 0.1% naphthylethylenediamine in 2% phosphoric acid) was added and the microplate was left to rest for 10 min. The microplate was read at 560 nm. The potassium phosphate buffer was used as a negative control and 2% phosphoric acid, instead of the Griess reagent, was added to the blanks. Ascorbic acid was used as a positive control. The experiments were made in triplicate in a Synergy™ HTX microplate reader.

##### Analysis of the Antioxidant Responses

The Weibull model was applied to estimate the half maximal effective concentration EC_50_ [48], with the mathematical modeling made at a 95% confidence level, after verifying the residual normality by a Shapiro–Wilk test with *p* > 0.05.

#### 3.3.3. Antimicrobial Activity

The antimicrobial capacity of the three optimal extracts was tested by the plate diffusion method [33,49,50] against five bacteria: two Gram-positive (*Staphylococcus aureus* and *Bacillus cereus*) and three Gram-negative bacteria (*Pseudomonas aeruginosa*, *Salmonella enteritidis*, and *Escherichia coli*). DMSO was used as a negative control in all tests. Lactic acid at 40% m/V was chosen as the positive control. This choice was made because lactic acid, as a fermentation product, is a natural antimicrobial agent in food, the extract was developed aiming for human consumption, and all bacteria assessed were foodborne pathogens.

The lyophilized extract was weighed and dissolved in dimethyl sulfoxide (DMSO) and brought up to 20 mg/mL final concentration. The extract was sterilized by filtration through a 0.20 µm syringe filter. Pre-inoculum of all microorganisms was cultured in Mueller–Hinton broth (MHB) overnight at 37 °C. After, the inoculum concentration was normalized to the 0.5 MacFarland (0.09 to 0.110 turbidity at 600 nm) standard by dilution in fresh MHB [50]. To assess the antimicrobial potential of the optimal extracts, Petri dishes containing Mueller–Hinton agar (MHA) were seeded with 100 µL of the microorganism under test and uniformly spread with sterile swabs. As a negative control, 15 µL of DMSO was applied on the dishes. As a positive control, 15 µL of 40% (*v*/*v*) lactic acid, corresponding to 7.2 mg of total mass applied, was added. Finally, 15 µL of extract at 20 mg/mL (0.3 mg of mass applied) was applied, which corresponds to ~24 times more diluted (*w*/*w*) than the positive control of lactic acid. All the samples were incubated at 37 °C for 24 h and the inhibition-zone diameter was measured with a digital caliper rule [51].

#### 3.3.4. Cholinesterase Inhibitory Capacity

The cholinesterase (AChE and BuChE) inhibition capacity was performed by the methodology first proposed by Ellman et al. [52] and is based on the measurement of the thiocholine released during the acetylcholine/butyrylcholine hydrolysis under the influence of acetylcholinesterase (AChE) or butyrylcholinesterase (BuChE), respectively. In a 96-well microplate, 25 µL of extract (1 mg/mL in the reaction volume), 125 µL of 3 mM DTNB, 25 µL of substrate ATCI or BTCI, and, finally, 50 µL of buffer (Tris-HCl; pH = 8) was added. The reaction kinetics was monitored by measuring the slopes of six absorbances, at 405 nm, for 1 min 44 s, after deducting the blanks (all reagents except the enzyme) [36]. The test was conducted using a Synergy™ HTX microplate reader.

## 4. Conclusions

Most carotenoids are used in industries, particularly food industries, because of the presence of important bioactivities with beneficial properties for human health. Among these carotenoids, Fx is a well-known carotenoid with several reported bioactivities as antioxidant, neuroprotective, and antimicrobial effects. Therefore, for its easier incorporation, future uses, and commercialization, it is important to understand its kinetic extraction behavior. In this study, Fx was obtained from *U. pinnatifida* using conventional heat extraction with ethanol and ethanolic solutions as solvent. To understand its extraction behavior, the experimental data obtained were fitted to the theoretical models, using a first-order kinetic equation, and structured in two parameters that allowed the extraction process to be understand in a much simpler form. The maximum concentrations of Fx that could be extracted (*k*) at the highest possible extractive rates (*r*) were the best conditions selected for each response. The optimal conditions selected for *Y*_1_ were *T* = 45 °C, *S* = 70%, and *t* = 66 min, obtaining ~5.240 mg Fx/g AS; for *Y*_2_ were *T* = 65 °C, *S* = 60%, and *t* = ~10 min, obtaining ~450 mg E/g AS; and for *Y*_3_ were *T* = 45 °C, *S* = 70%, and *t* = 45 min, obtaining ~12.3 mg Fx/g E. Furthermore, their values of RMSE-MAE were low and their *R*^2^ were excellent (above 0.950), meaning that the experimental data were a good fit to the kinetic equation used. In addition, the bioactive analysis revealed the presence of different pigments, phenolics, and flavonoids. The three optimums displayed biological properties, highlighting a remarkable antimicrobial activity against different bacterial strains. Also, the extracts displayed antioxidant activities and inhibited cholinesterase activity. These results could have potential uses for the food, cosmetic, or pharmaceutical industries, helping them to reduce costs associated with energy and solvent consumption, lower their environmental impact, and maximize the extraction yield of Fx, for future applications.

## Figures and Tables

**Figure 1 marinedrugs-21-00414-f001:**
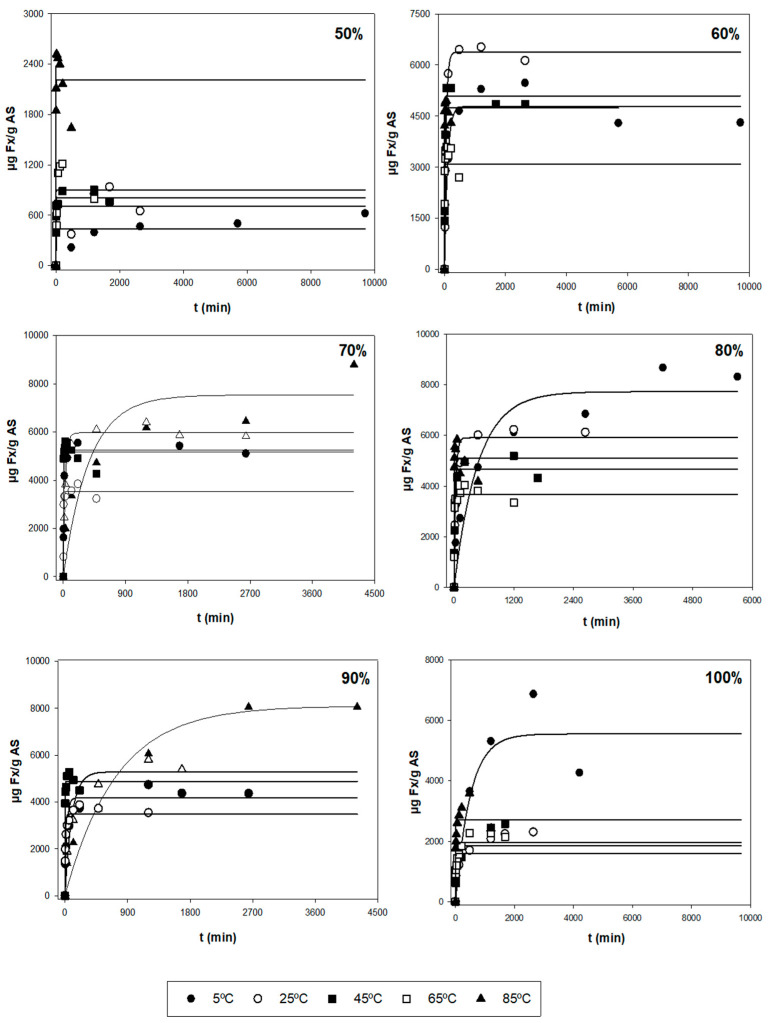
Representation of experimental data of the amount of Fx (*Y*_1_, symbols, µg Fx/g AS) and model-predicted data (lines) as a function of time (*t*, min) using Equation (1) for each of the other variables tested (*T* and *S*).

**Figure 2 marinedrugs-21-00414-f002:**
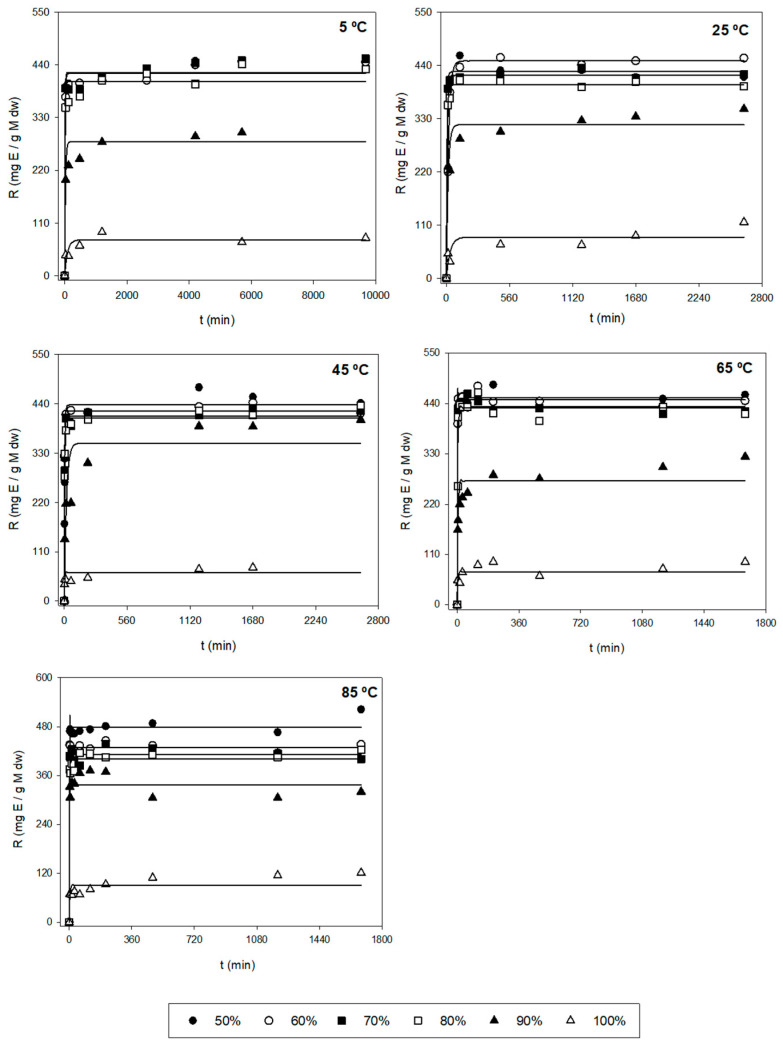
Representation of experimental data of the amount of E (*Y*_2_, symbols, mg E/g AS) and model-predicted data (lines) as a function of time (*t*, min) using Equation (1) for each of the other variables tested (*T* and *S*).

**Figure 3 marinedrugs-21-00414-f003:**
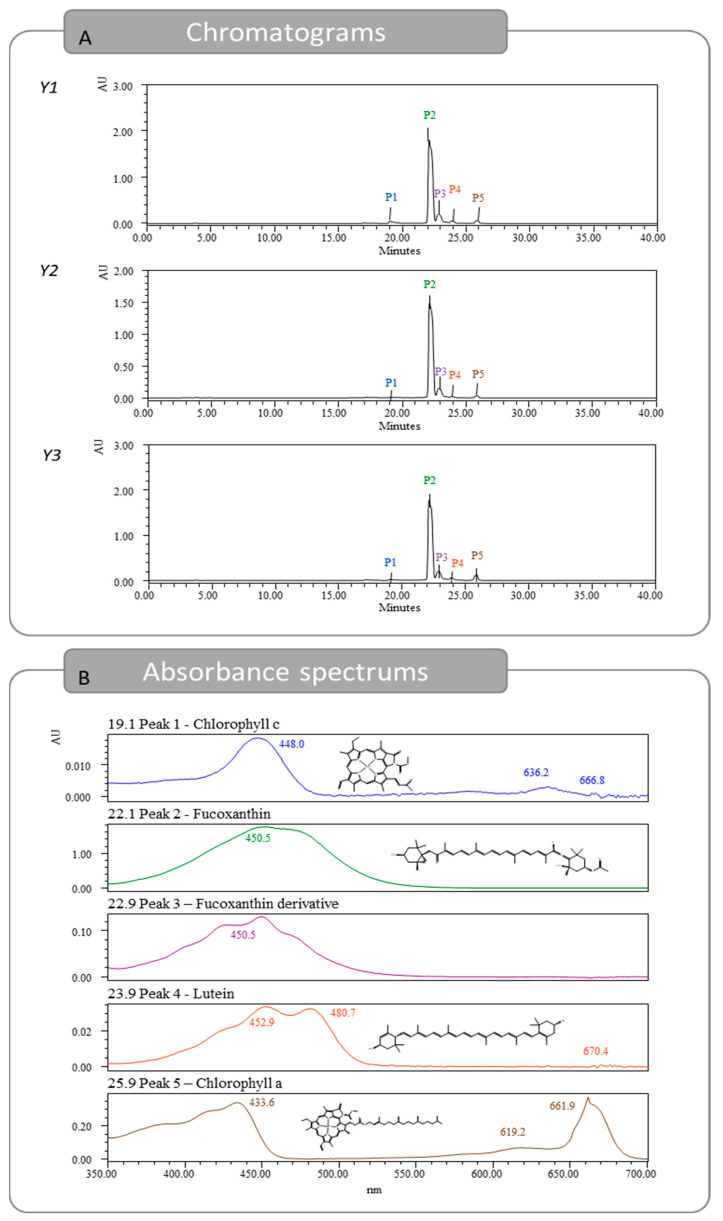
Chromatograms obtained for responses *Y*_1_, *Y*_2_, and *Y*_3_ from HPLC-DAD and absorption spectrums for the five identified pigments found in *Y*_1_.

**Figure 4 marinedrugs-21-00414-f004:**
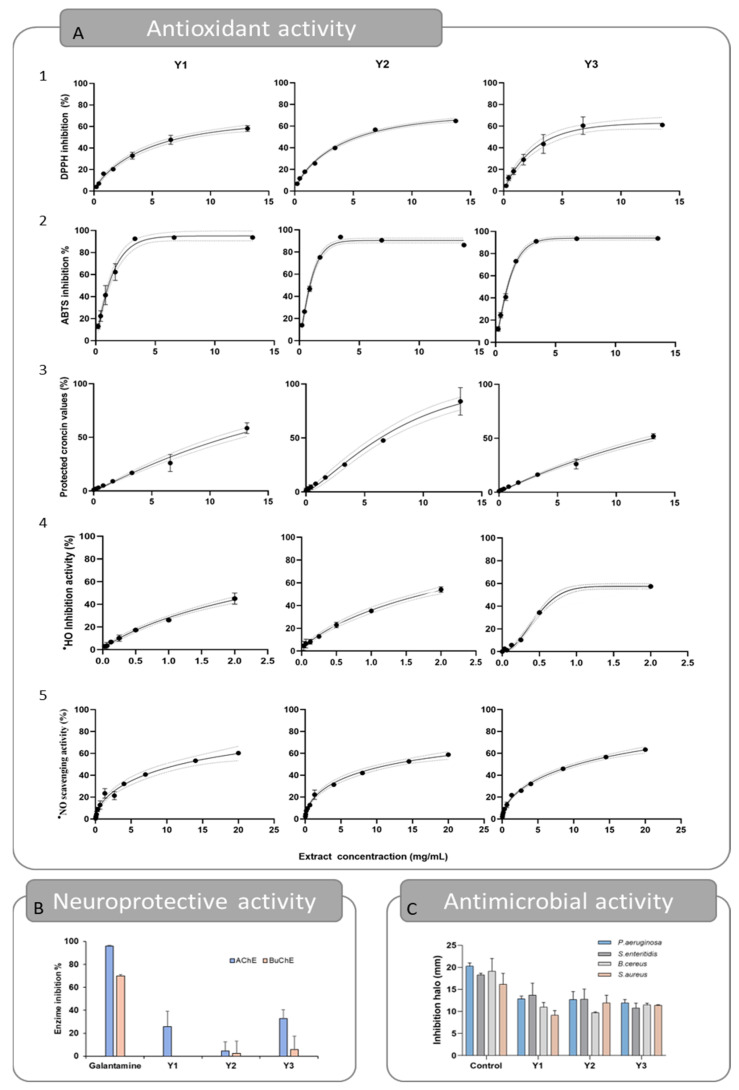
Representation of experimental data obtained for responses *Y*_1_, *Y*_2_, and *Y*_3_ for the antioxidant (**A**) (A1—Scavenging activity of the DPPH radical, A2—Scavenging activity of the ABTS^•+^ radical, A3—Crocin bleaching assay, A4—Scavenging activity of the hydroxyl radical, A5—Scavenging activity of the NO radical), neuroprotective (**B**), and antimicrobial (**C**) activities.

**Table 1 marinedrugs-21-00414-t001:** Kinetic results of Fx extraction as a function of solvent concentration (*S*, %), time (*t*, min), and extraction temperature (*T*, °C). Responses are expressed as: *Y*_1_, fucoxanthin per gram of alga sample (mg Fx/g AS dw); *Y*_2_, extract yield or mg extract per gram of alga sample (mg E/g AS dw).

Extraction Variables		Extraction Responses for Ethanol Solvent in Different Concentrations											
		*Y*_1_ (mg Fx/g AS dw)						*Y*_2_ (mg E/g AS dw)					
T (°C)	t (min)	50%	60%	70%	80%	90%	100%	50%	60%	70%	80%	90%	100%
5	30	0.71	1.90	1.99	1.76	1.38	1.08	394.4	372.5	390.8	351.1	199.7	42.7
	120	0.52	3.23	3.35	2.73	2.26	1.47	392.4	400.4	388.8	362.4	230.3	41.1
	480	0.22	4.66	4.72	4.74	3.69	3.66	393.8	401.8	390.2	374.3	243.6	63.3
	1200	0.40	5.29	6.18	6.13	6.06	5.31	415.8	408.4	414.8	408.2	278.9	90.7
	2640	0.47	5.47	6.44	6.86	8.06	6.87	425.1	407.7	432.1	422.2	357.5	123.3
	4200	0.38	3.61	8.79	8.68	8.06	4.27	447.9	439.9	443.6	399.5	291.0	57.7
	5700	0.50	4.30	8.79	8.32	8.48	0.74	443.9	447.1	449.0	442.1	299.0	69.6
	9680	0.62	4.31	7.19	10.06	7.81	5.80	445.9	445.8	452.9	430.8	330.2	78.9
25	15	0.76	1.24	2.45	2.45	1.47	0.86	394.0	219.9	391.5	357.9	231.0	51.1
	30	0.44	2.90	3.82	3.26	1.88	1.12	404.5	383.9	409.0	371.9	222.9	34.8
	120	1.82	5.74	5.49	4.93	3.22	1.22	460.4	436.3	414.8	409.1	287.9	168.3
	480	0.37	6.45	6.10	6.02	4.76	1.70	428.9	456.0	419.5	406.7	303.0	69.6
	1200	0.85	6.53	6.40	6.23	5.81	2.09	430.1	440.9	433.5	396.3	326.3	68.5
	1680	0.94	5.98	5.87	5.42	5.39	2.25	415.0	449.1	412.5	405.6	334.4	88.2
	2640	0.65	6.13	5.82	6.13	6.68	2.31	416.1	454.9	420.7	397.4	349.5	116.1
45	3	0.39	1.42	1.62	1.35	0.15	0.61	172.0	323.3	321.1	278.4	2.7	36.6
	5	0.59	1.71	1.98	2.24	1.36	0.86	264.6	319.3	291.1	329.0	137.8	48.6
	15	0.71	3.95	4.18	3.35	2.08	1.01	430.6	415.9	407.7	381.0	217.1	46.6
	60	0.73	5.32	4.92	4.34	2.98	1.20	398.0	425.2	389.7	393.7	219.1	43.9
	210	0.89	5.32	5.55	4.96	3.73	1.48	420.6	419.9	419.7	403.7	308.3	51.9
	1200	0.90	4.54	4.69	5.20	4.74	2.44	476.0	433.2	414.3	423.6	389.5	70.6
	1680	0.77	4.86	5.42	4.32	4.37	2.58	455.3	441.9	430.3	413.6	388.9	74.6
	2640	1.17	4.85	5.11	3.69	4.36	2.89	441.3	419.2	424.3	435.0	404.2	90.5
65	3	1.07	1.92	0.82	1.20	1.48	0.93	433.9	395.7	424.2	259.1	164.3	53.9
	5	1.31	2.89	2.99	0.57	1.98	1.04	464.6	450.3	424.2	409.9	185.0	53.9
	15	0.48	3.25	3.33	3.16	2.62	1.05	428.6	441.6	430.8	436.5	219.5	47.9
	30	0.63	3.49	3.33	3.51	3.00	1.19	455.9	453.6	444.1	432.5	234.8	70.6
	60	1.10	3.59	3.59	3.45	3.22	1.43	441.9	431.6	461.4	431.9	244.8	45.3
	120	1.18	3.34	3.57	3.72	3.65	1.56	451.9	477.6	444.8	463.7	381.9	86.5
	210	1.22	3.55	3.84	4.04	3.87	1.84	480.6	443.6	422.9	419.2	284.1	93.9
	480	0.32	2.70	3.24	3.82	3.72	2.27	413.9	445.0	429.5	402.0	276.1	63.2
	1200	0.79	2.66	2.92	3.34	3.54	2.27	449.9	433.6	416.9	433.2	301.4	78.6
85	3	1.85	4.22	4.89	4.75	3.95	1.77	469.2	435.2	406.9	374.6	331.6	68.6
	5	2.11	4.65	5.19	5.12	4.44	1.78	473.2	433.2	404.2	364.6	305.6	67.9
	15	2.52	4.87	5.34	5.55	4.63	1.99	462.6	409.9	423.6	389.3	342.3	66.6
	30	2.51	4.92	5.62	5.45	5.11	2.24	462.6	423.2	396.9	370.6	340.3	75.9
	60	2.47	4.96	5.56	5.85	5.28	2.61	469.2	432.6	384.3	415.3	365.6	67.9
	120	2.40	4.62	5.25	4.50	4.93	2.86	473.2	425.2	409.6	412.6	371.6	80.6
	210	2.16	4.31	4.90	5.00	4.50	3.12	480.6	444.5	436.9	403.9	368.9	93.2
	480	1.64	3.58	4.27	4.18	3.79	3.60	487.9	432.6	425.6	411.3	305.0	108.8
	1200	1.00	2.38	3.21	3.15	2.46	2.43	465.9	415.2	412.9	404.6	305.0	114.6

**Table 2 marinedrugs-21-00414-t002:** The parameters *k* (µg/g) and *r* (min^−1^) values at different temperatures (*T*) and ethanol concentrations (*S*). In addition, *R*^2^ values for different temperatures and ethanol proportions are shown.

VARIABLES		*Y* _1_					*Y* _2_					*Y* _3_				
*T*	*S*	*k_Y*1*_*		*r_Y*1*_*		*R* ^2^	*k_Y*2*_*		*r_Y*2*_*		*R* ^2^	*k_Y*3*_*		*r_Y*3*_*		*R* ^2^
(°C)	(%)	(µg Fx/g AS)		(min^−1^)			(mg E/g AS)		(min^−1^)			(mg Fx/g E)		(min^−1^)		
5	50	440	±330.6	0.089	±0.001	0.6439	424	±13.2	0.089	±0.042	0.9787	1.1	ns	0.094	ns	0.4885
	60	4787	±332.3	0.012	±0.004	0.9825	422	±13.2	0.072	±0.025	0.9825	10.8	±2.1	0.016	ns	0.9825
	70	7533	±348.5	0.003	±0.001	0.9731	424	±13.2	0.084	±0.037	0.9731	16.6	±2.3	0.005	±0.004	0.9731
	80	7737	±414.2	0.002	±0.001	0.9657	406	±13.2	0.067	±0.022	0.9657	19.4	±2.5	0.002	±0.002	0.9657
	90	8044	±582.3	0.001	±0.001	0.9441	269	±15.8	0.044	±0.017	0.9441	24.0	±3.3	0.003	±0.002	0.9441
	100	5554	±477.1	0.002	±0.001	0.8202	75	±17.6	0.011	ns	0.8202	61.4	±2.7	0.009	±0.002	0.8202
25	50	704	±369.6	0.165	±0.001	0.6771	427	±14.7	0.165	±0.070	0.9887	2.0	ns	0.159	ns	0.3498
	60	6288	±358.0	0.019	±0.005	0.9912	449	±15.6	0.052	±0.009	0.9912	13.9	±2.4	0.029	±0.025	0.9912
	70	5964	±338.9	0.034	±0.009	0.9978	419	±14.6	0.179	±0.089	0.9978	14.2	±2.4	0.037	±0.030	0.9978
	80	5912	±390.1	0.028	±0.008	0.9948	400	±14.8	0.143	±0.053	0.9948	14.3	±2.4	0.035	±0.028	0.9948
	90	5286	±434.6	0.010	±0.004	0.9341	317	±15.5	0.064	±0.017	0.9341	16.6	±2.5	0.020	±0.015	0.9341
	100	1843	±384.5	0.031	±0.029	0.7484	85	±17.5	0.030	±0.028	0.7484	23.5	±2.5	0.132	±0.117	0.7484
45	50	808	±344.4	0.232	ns	0.9550	438	±15.6	0.177	±0.031	0.9811	2.1	ns	0.269	ns	0.5246
	60	5018	±369.5	0.098	±0.031	0.9787	425	±15.7	0.382	±0.076	0.9787	11.6	±2.6	0.133	±0.129	0.9787
	70	5254	±367.8	0.105	±0.032	0.9594	409	±15.7	0.377	±0.078	0.9594	12.3	±2.5	0.155	±0.141	0.9594
	80	4676	±367.6	0.105	±0.036	0.9862	403	±15.7	0.368	±0.076	0.9862	11.2	±2.5	0.171	±0.171	0.9862
	90	4141	±411.6	0.037	±0.017	0.8638	333	±18.2	0.060	±0.015	0.8638	11.9	±2.4	0.235	±0.222	0.8638
	100	1962	±387.6	0.058	±0.052	0.7838	58	±15.8	0.339	ns	0.7838	30.1	±2.5	0.185	±0.069	0.7838
65	50	900	±330.6	1.062	ns	0.5905	453	±13.2	1.062	±0.664	0.9916	2.0	±2.0	1.164	ns	0.4409
	60	3093	±279.4	5.320	ns	0.9909	450	±13.9	0.733	±0.241	0.9909	7.4	±2.1	0.403	ns	0.9909
	70	3520	±350.6	0.205	±0.093	0.9906	437	±13.6	1.120	±0.825	0.9906	8.0	±2.2	0.216	ns	0.9906
	80	3661	±308.3	0.131	±0.074	0.9708	433	±14.3	0.376	±0.071	0.9708	8.8	±2.5	0.100	ns	0.9708
	90	3458	±330.5	0.146	±0.067	0.8919	261	±14.6	0.276	±0.080	0.8919	12.3	±2.1	0.439	ns	0.8919
	100	1591	±316.5	0.207	±0.204	0.7444	74	±14.4	0.335	ns	0.7444	24.0	±2.1	0.377	±0.194	0.7444
85	50	2209	±280.7	1.951	ns	0.8523	478	±11.8	1.331	ns	0.9868	4.1	±1.8	1.308	ns	0.4123
	60	4516	±279.4	7.167	ns	0.9942	429	±11.7	7.167	ns	0.9942	9.3	±1.7	7.156	ns	0.9942
	70	5129	±280.3	2.146	ns	0.9864	411	±11.8	1.533	ns	0.9864	11.3	±1.7	1.813	ns	0.9864
	80	5096	±289.8	0.910	±0.786	0.9800	401	±12.1	0.798	±0.326	0.9800	11.5	±1.8	1.580	ns	0.9800
	90	4690	±301.8	5.061	ns	0.9417	337	±11.9	1.176	ns	0.9417	12.0	±1.8	1.452	ns	0.9417
	100	2711	±286.3	0.253	±0.146	0.6906	91	±12.4	0.363	±0.318	0.6906	29.1	±1.8	0.648	±0.410	0.6906

Abbreviations: ns: not statistically significant; *R*^2^: coefficient of determination.

**Table 3 marinedrugs-21-00414-t003:** Trend and dependence of the kinetic parameters (*k* and *r*) with *T* and *S*.

Responses	Constants	Kinetic Parameters	Trend	Dependence
*Y*_1_ (µg Fx/g AS)	Same solvent	*k*	constant	not depend on *T*
		*r*	increases with *T*	depend on *T*
	Same temperature	*k*	increases and decreases with *S* (curve)	depend on *S*
		*r*	constant	not depend on *S*
*Y*_2_ (mg E/g AS)	Same solvent	*k*	constant	not depend on *T*
		*r*	increases with *T*	depend on *T*
	Same temperature	*k*	decreases with *S*	depend on *S*
		*r*	constant	not depend on *S*
*Y*_3_ (mg Fx/g E)	Same solvent	*k*	constant	not depend on *T*
		*r*	increases with *T*	depend on *T*
	Same temperature	*k*	constant	not depend on *S*
		*r*	constant	not depend on *S*

**Table 4 marinedrugs-21-00414-t004:** Results obtained from the bioactive analysis of the optimal values *Y*_1_, *Y*_2_, and *Y*_3_.

	*Y* _1_	*Y* _2_	*Y* _3_	Control
HPLC-DAD analysis (mg/g dw)				
Chl c	0.17	0.02	0.04	Chl a standard
Fx	4.53	3.82	4.42	Fx standard
Fx derivative	0.24	0.27	0.26	Fx standard
Lutein	0.02	uql.	0.01	Lutein standard
Chl a	0.14	0.07	0.25	Chl a standard
Analysis of secondary metabolites				
TPC (mg GAE/g dw)	3.43 ± 0.40	3.26 ± 1.35	2.04 ± 0.20	-
TFC (µg QE/g dw)	90.47 ± 5.45	54.17 ± 3.17	86.20 ± 5.95	-
Antioxidant activity				
				Quercetin
DPPH (EC_50_ mg/mL)	3.15 ± 0.83	2.67 ± 0.80	1.82 ± 0.95	0.80 ± 0.04
ABTS•+ (EC_50_ mg/mL)	1.00 ± 1.09	2.67 ± 0.80	0.92 ± 0.92	0.14 ± 0.01
Crocin (EC_50_ mg/mL)	11.52 ± 1.07	7.01 ± 1.09	13.13 ± 0.10	6.67 ± 0.33
				Ascorbic acid
●HO radical (EC_50_ mg/mL)	2.00 ± 0.84	1.76 ± 0.78	0.44 ± 1.93	0.45 ± 0.02
●NO radical (EC_50_ mg/mL)	11.76 ± 0.54	12.62 ± 0.53	10.48 ± 0.58	0.18 ± 0.01
Cholinesterase inhibitory activity				
				Galantamine
AChE (inhibition %)	25.75 ± 13.31	4.54 ± 8.06	32.85 ± 7.46	95.85 ± 0.82
BuChE (inhibition %)	-	2.32 ± 10.79	5.72 ± 11.57	69.78 ± 1.43
Antimicrobial activity (halo inhibition in mm)				
				Lactic acid
*S. aureus*	9.14 ± 1.05	11.94 ± 1.70	11.40 ± 0.09	16.15 ± 2.46
*B. cereus*	10.97 ± 1.06	9.73 ± 0.19	11.46 ± 0.40	19.09 ± 2.93
*P. aeruginosa*	12.84 ± 0.62	12.69 ± 1.79	11.96 ± 0.75	20.28 ± 0.72
*S. enteritidis*	13.70 ± 2.70	12.76 ± 2.32	10.78 ± 1.11	18.25 ± 0.42
*E. coli*	-	-	-	17.82 ± 0.56

Fx, fucoxanthin; Chl. chlorophyll; *Y*_1_, *Y*_2_, and *Y*_3_, three optimal responses; uql., under quantification limit; TPC, total phenolic compounds; GAE, gallic acid equivalents; TFC, total flavonoid content; QE, quercetin equivalents; EC_50_, half maximal effective concentration; DPPH, 2,2-diphenylpicrylhydrazyl assay; ABTS, 2,2′-azino-bis(3-ethylbenzothiazoline-6-sulfonic acid) assay; HO^●^, radical hydroxyl scavenging activity; NO^●^, inhibition of nitric oxide radical production assay; AChE, acetylcholinesterase; BuChE, butyrylcholinesterase; *P. aeruginosa* (*Pseudomonas aeruginosa*); *S. aureus* (*Staphylococcus aureus*); *S. enteritidis* (*Salmonella enterica*); *B. cereus* (*Bacillus cereus*); *E. coli* (*Escherichia coli*).

## Data Availability

The data presented in this study are available on request from the corresponding author.

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
