# Peer review of "Kinetic Extraction of Fucoxanthin from Undaria pinnatifida Using Ethanol as a Solvent"

_marinedrugs, 2023, doi:10.3390/md21070414_

Round 1

Reviewer 1 Report

I have evaluated the manuscript by Catarina Lourenço-Lopes et al. I have the following recommendations and comments prior to acceptance of the manuscript:

1.  In Introduction: Another green Natural Deep Eutectic Solvents were recently proposed for extraction of fucoxanthin.

2. In Section 2: please justify rationality of selection of parameters used in experiments and diapasons in which parameters were varied.

3. EtOH is very volatile. The temperature 85 °C  and EtOH concentration over 70% at such high temperature not rational.

4. How authors have prepared EtOH 100%?

5. In Table 1 authors have not provided statistical data. In how many repetitions were the experiments carried out?

6. Please indicate if results presented in Table 1 were statistically significant different.

7. Fig. 1 abd 2  - no statistics.

8. Table 2 - see my comments 5 and 6.

9. No statistics in Fig. 4A

10. The antimicrobial activity should be presented as MIC for correct comparison.

11. Some figures duplicate the data from tables. Please revise (for exa,ple Cholinesterase inhibition, antimicrobial activity, etc).

12. Sect. 3.1: who have identified seaweed?

13. In case of industrial application of proposed technology  a huge amount of EtOH 70%  will be used. As authors suggested 30 g seaweed should be extracted with 1 L of EtOH. It mean 30 kg seaweed should be extracted in 1000 L.L of EtOH.  It seems not rational.

14. The conclusion should be based on statistically proven results.

The language is OK, just polishing is recommended.

Author Response

I have evaluated the manuscript by Catarina Lourenço-Lopes et al. I have the following recommendations and comments prior to acceptance of the manuscript:

  1. In Introduction: Another green Natural Deep Eutectic Solvents were recently proposed for extraction of fucoxanthin.

Response: The reviewer is right we have not introduced proper other alternative solvents. To fix this issue, the introduction has been reformulated and the recent application of another natural deep eutectic solvent for the extraction of fucoxanthin has been incorporated into the manuscript accompanied by two references to support this information:

“Regarding the selection of the extractive solvents, some of the most recent approaches include the use of bio-solvents, ionic liquids, or deep eutectic solvents as greener and less toxic options. However, they have been less explored than conventional solvents due to their straightforward application with simpler equipment and low initial cost requirements [10–12]. Among the conventional organic solvents, those polar (acetone, ethanol, or ethyl acetate) are much preferred to recover Fx due to their polar nature. Ethanol has been demonstrated to provide efficient results, especially when used at high temperatures [11,12].”

  1. In Section 2: please justify rationality of selection of parameters used in experiments and diapasons in which parameters were varied.

Response: The reviewer is correct, no explanation of parameter selection was added to the manuscript. For a kinetic study, as established in the manuscript, is necessary to obtain a minimum number of sample points, in the present case the target number was 258 points. To reach this number it was necessary to perform a selection of the incubation times to test. Based on our previous experience, and as proved by the values displayed in table 1, short incubations for lower temperatures need to be discarded as well as long times for high temperatures. In this way 65 and 85°C were selected times from 5 to 1200 min, for 45 °C shorter times were maintained while intermediate values such as 30 or 480 min were removed to incorporate data of longer incubation times (1680 and 2640 min). These longer times were kept for further experiments based on 25°C and the shortest incubation time was 15 min. Finally, for the lowest temperature, the shortest incubation time was 30 min and the maximal was 7 days (9680 min). To briefly explain this point, the following sentence was included in the text: “As observed in these experimental combinations, higher temperatures were tested at shorter incubation times while lower temperatures were assessed up to the longest times”.

  1. EtOH is very volatile. The temperature 85 °C and EtOH concentration over 70% at such high temperature not rational.

Response: Indeed, the ebullition temperature of pure ethanol is 78ºC. We had in consideration this information when designing the experiment. That is why we maintained all extraction vessels sealed during extraction and cool down times to avoid the evaporation of the solvent and influence the results obtained. To clarify this point, it was added the following sentence “Ambar bottles used for the extraction were sealed during incubation and cool down times to prevent evaporation issues.”

  1. How authors have prepared EtOH 100%?

Response: The reviewer is right, the ethanol used was bought at the higher purity available, 96°, 100% was an approximation that was not explained in the manuscript. We have introduced a sentence in material and methods to clarify this point.

“Even though the initial concentration of ethanol was 96° we considered it as 100% to provide round numbers of the following aqueous solutions.”

  1. In Table 1 authors have not provided statistical data. In how many repetitions were the experiments carried out?
  2. Please indicate if results presented in Table 1 were statistically significant different.
  3. Fig. 1 and 2 - no statistics.
  4. Table 2 - see my comments 5 and 6.
  5. No statistics in Fig. 4A

Response: In ordinary studies, statistical data is provided since punctual measurements are developed and so compared. However, the present document displays a kinetic study for which several parameters are assessed along a continuous variation of conditions. This kind of models permit to evaluate a range of optimal conditions which is later compared to the model-predicted results (figures 1 and 2). As explained in the manuscript “Table 2 shows the parametric values k and r with their 95% confidence intervals for three responses (Y1, Y2, and Y3) at different temperatures (T) and ethanol concentrations (S), found by the tool Solver”. While Table 3 presents the dependence of parameters k and r against temperature and ethanol solvent.

  1. The antimicrobial activity should be presented as MIC for correct comparison.

Response: The method employed to determine the antimicrobial activity has been used and published in several research papers (including 10.1016/j.foodchem.2014.09.102; 10.3390/foods10081915; 10.3390/antiox12020435). Assessing antimicrobial capacity by observing and measuring zones of inhibition is a common method applied to various natural matrices, such as in this review (10.3390/antibiotics9100712).

  1. Some figures duplicate the data from tables. Please revise (for exa,ple Cholinesterase inhibition, antimicrobial activity, etc).

Response: We realize some data is duplicated, however, since the manuscript presents a kinetic study, we consider key to present data but also to present plots to provide further visualization of the results. Indeed, Figure 4 helps to analyze the results summarized in Table 4 providing a more accessible interpretation of the results in a graphical manner. We would like to keep both if that is okay with the reviewer.

  1. Sect. 3.1: who have identified seaweed?

Response: The algae material used was supplied by a company (AlgaMar) established in 1996. They collect algae manually on the Galician coastline. They have more than 20 years of experience in the sector. Regarding, species identification, we are aware that a biologist with identification skills works as part of the team. Besides, two biologists are currently working as part of our laboratory and double check the identification of the species.

  1. In case of industrial application of proposed technology a huge amount of EtOH 70% will be used. As authors suggested 30 g seaweed should be extracted with 1 L of EtOH. It mean 30 kg seaweed should be extracted in 1000 L.L of EtOH. It seems not rational.

Response: The reviewer is right, the scale-up to industrial sectors would require a final optimization of the S:L ratio. The current S:L ratio was selected to develop all the experimental work at the lab scale. A sentence remarking on this point was introduced in the text: “In this sense, the solid:liquid (S:L) ratio may also be adjusted to maximize the use of the solvent, since the S:L suggested at the present work is hardly scalable at industrial sectors.”

  1. The conclusion should be based on statistically proven results.

Response: Conclusions are exposed based on how experimental data fitted to theoretical models.

Reviewer 2 Report

In this paper “Undaria pinnatifida as a source of fucoxanthin: kinetics extraction using ethanol as a solvent”, the aim was to study the kinetic behavior of the Fx molecule and optimize three variables, time (t), temperature (T) and solvent concentration (S), to obtain the best Fx yield from U. pinnatifida using a heat-assisted extraction. In addition, for the selected optimums, bioactive compounds, and several properties (antioxidant, antimicrobial and cholinesterase inhibitory activities) were evaluated to estimate the biological potential of the extracts, which could be interesting for industrial applications.

In my opinion, I found the paper well summarized, structured and written but I also recognize that it is a highly studied research topic (“bioactive compounds extraction from microalgae or algae”).

For publishing this manuscript in Marine Drugs in its current form, there are several sentences and suggestions where improvement is needed for conveying the message of this work to the scientific community. Nonetheless, I think the manuscript will generally be of interest to the industrial applications. I have a few comments and suggestions:

Firstly, I recommend changing the units of Fx from µg/g to mg/g, so the numbers will be smaller in the Table 1 is more readable.

In the material and methods section, Total Phenolic Content (TPC) and Total Flavonoid Content (TFC) should be described by what standard was used. For example, Gallic acid and quercetin, respectively, and their concentration for the standard curve (only mentioned in Table 4)

Why not compare it to some study that uses green techniques? That is supercritical fluids, liquid pressurized, etc. Many reports evaluate the fucoxanthin extraction. It would be interesting why you decide the solvent extraction versus other innovative techniques.

And my main question is, why decide to use this conventional technique versus other very efficient clean ones? In the first part of your study, the extraction time is up to 9680 min, explain better.

Finally, the presence of bioactive, neuroprotective, and antioxidant compounds in the extract is very relevant for numerous applications. Therefore, these values justify the richness of these extracts (not only in fucoxanthin) and the use of Undaria pinnatifida for food applications.

Author Response

In this paper “Undaria pinnatifida as a source of fucoxanthin: kinetics extraction using ethanol as a solvent”, the aim was to study the kinetic behavior of the Fx molecule and optimize three variables, time (t), temperature (T) and solvent concentration (S), to obtain the best Fx yield from U. pinnatifida using a heat-assisted extraction. In addition, for the selected optimums, bioactive compounds, and several properties (antioxidant, antimicrobial and cholinesterase inhibitory activities) were evaluated to estimate the biological potential of the extracts, which could be interesting for industrial applications.

In my opinion, I found the paper well summarized, structured and written but I also recognize that it is a highly studied research topic (“bioactive compounds extraction from microalgae or algae”). For publishing this manuscript in Marine Drugs in its current form, there are several sentences and suggestions where improvement is needed for conveying the message of this work to the scientific community. Nonetheless, I think the manuscript will generally be of interest to the industrial applications. I have a few comments and suggestions:

The authors would like to thank the reviewer for the time and attention dedicated to the manuscript and for the positive comments.

  1. Firstly, I recommend changing the units of Fx from µg/g to mg/g, so the numbers will be smaller in the Table 1 is more readable.

Response: Table 1 was updated for mg/g as requested.

  1. In the material and methods section, Total Phenolic Content (TPC) and Total Flavonoid Content (TFC) should be described by what standard was used. For example, Gallic acid and quercetin, respectively, and their concentration for the standard curve (only mentioned in Table 4)

Response: The information was added in the material and methods section.

  1. Why not compare it to some study that uses green techniques? That is supercritical fluids, liquid pressurized, etc. Many reports evaluate the fucoxanthin extraction. It would be interesting why you decide the solvent extraction versus other innovative techniques.

Response: We completely agree with the reviewer. We have introduced the utilization of greener and more innovative extraction techniques in the introduction.

We also have reformulated the introduction and the first part of “results and discussion” to explain the selection of ethanol as a solvent for the extractions and briefly compare our results to other techniques.

“These values were in most cases higher than most of those reported in previous studies. For instance, other maceration studies provided concentrations of 0.7 mg Fx/g of dw when using EtOH or 2.67 mg Fx/g of dw when using methanol. Microwave-assisted extraction provided values of 0.99 mg Fx/g of dw whereas supercritical fluid extraction using CO2 with EtOH as co-solvent [4,12].”

  1. And my main question is, why decide to use this conventional technique versus other very efficient clean ones? In the first part of your study, the extraction time is up to 9680 min,

Response: As explained previously, we agree with the reviewer and we have also applied green extraction technologies for the extraction of fucoxanthin. Nevertheless, and as explained in the introduction, the optimization of maceration extraction was also necessary as many times industries prefer to use simple and easy-to-operate methodologies, as the initial investment in equipment can be a lot smaller for this kind of protocol. Also, the optimal extraction conditions obtained helped us to design the experiments for the MAE and UAE extraction, so in that sense, this work was used as a preliminary study for future analysis. We extended the extraction time up to 7 days because there were many different parameters used in the literature and we wanted to have the complete range of times. Nevertheless, as stated in the manuscript we reject this option as a reliable approach, instead we pointed out the use of much shorter incubation times.

In addition to the comparison of our results with other previously published where no better recovery results were displayed, we tried to clarify the election of ethanol as a solvent extract, we have discussed the reasons for this selection in the first paragraph of “Results and discussion”:

“For the choice of the extraction solvent, many factors must be taken into consideration, such as their melting and boiling point, their polarity, density and viscosity [11]. Ethanol is an excellent choice because it has a large range between its melting and boiling points (-114 to 78°C), which allows a large range of extraction temperatures; it has a relatively low viscosity (1.2 cP), providing a more favorable mass transfer from the matrix to the solvent due to its capacity to easily penetrate solid samples; and has a medium polarity (4.4), which is great to extract compounds like Fx that can be dissolved in mid-polar solvent systems [13]. However, water and ethanolic mixtures with more than 60% water were found to be ineffective in solubilizing Fx [14]. Besides, ethanol is considered to have a low environmental impact, low toxicity and is therefore considered a safe and green solvent to be used in the food industry. Hereby, ethanol or ethanolic solutions are great solvents for the extraction of Fx and its future incorporation in food, cosmetic or pharmaceutical products. Therefore, the present kinetic study was performed using ethanol as an organic solvent to recover Fx.”

  1. Finally, the presence of bioactive, neuroprotective, and antioxidant compounds in the extract is very relevant for numerous applications. Therefore, these values justify the richness of these extracts (not only in fucoxanthin) and the use of Undaria pinnatifida for food applications.

Response: We appreciate the positive input made by the reviewer.

Round 2

Reviewer 1 Report

Authors have revised the manuscript. Hovewer, some updates still required.

1. The application of Natural Deep Eutectic Solvents for the extraction of fucoxanthin was reported in next experimental articles: https://doi.org/10.3390/molecules26144198; https://doi.org/10.1016/j.jece.2022.108370

2. The phrase 'Even though the initial concentration of ethanol was 96° we considered it as 100% to provide round numbers of the following aqueous solutions.' is incorrect. please indicate exact concentration of EtOH used. Otherwise it is confusing if other concentrations provided in the manuscript must be recalculated taking in account that 100% is 96%, then 70% will be 67.2% (70*96/100)?

3. In how many repetitions were the experiments on the kinetic study?

4. Some figures duplicate the data from tables. Please revise (for exa,ple Cholinesterase inhibition, antimicrobial activity, etc). Authors should select most appropriate method for results presentation. Otherwise this is just increase in volume of paper without scinetific valus.

5. Sect. 3.1: who have identified seaweed? Please include this important information in the text of manuscript.

6. The authors' comments on my question 13 from the previous peer review round should be included in the manuscript as a limitation of this study.

Minor editing of English language required

Author Response

The authors have revised the manuscript. However, some updates still required.

  1. The application of Natural Deep Eutectic Solvents for the extraction of fucoxanthin was reported in next experimental articles: https://doi.org/10.3390/molecules26144198; https://doi.org/10.1016/j.jece.2022.108370

Response: We truly appreciate the specific information in relation to the use of NADES to recover fucoxanthin. We have used it to complete our introduction: “Among some of the natural deep eutectic solvents used for Fx recoveries can be underline mixtures of thymol with octa-, deca- or dodecanoic acids or lactic acid mixed with choline chloride or aqueous glucose [10,11]”. However, we do not consider it relevant to discuss these citations in the results and discussion section since they applied the methods to two different algae species (Tisochrysis lutea and Fucus vesiculosus) than the one we used Undaria pinnatifida.

  1. The phrase 'Even though the initial concentration of ethanol was 96° we considered it as 100% to provide round numbers of the following aqueous solutions.' is incorrect. please indicate exact concentration of EtOH used. Otherwise it is confusing if other concentrations provided in the manuscript must be recalculated taking in account that 100% is 96%, then 70% will be 67.2% (70*96/100)?

Response: We disagree with the reviewer at this point. We rather prefer to use round numbers. It is scientifically correct to use them. If any person rounds these numbers (67.2%) it results in the values we provide in our manuscript (70%). To try to better clarify this point we have rephrased the sentence as follows: “The initial concentration of ethanol was 96°, however, to facilitate data representation we labeled it as 100% and so the following aqueous solutions (50, 60, 70, 80 and 90%) whose real ethanol concentration is 48%, 57.6%, 67.2%, 76.8% and 86.4%, respectively.”

  1. In how many repetitions were the experiments on the kinetic study?

Response: Kinetic studies as well as RSM studies do not require repetitions of the conditions since it analyzes the total data cluster to compare the theoretical estimations against the experimental values obtained and understand if the mathematical model fit the real results.

  1. Some figures duplicate the data from tables. Please revise (for exa,ple Cholinesterase inhibition, antimicrobial activity, etc). Authors should select most appropriate method for results presentation. Otherwise this is just increase in volume of paper without scinetific valus.

Response: As we previously stated, we consider key to present numerical and graphical data to provide a better interpretation of the results. To prevent the excessive extension of the manuscript we already compacted all data and graphs into just one table and figure, respectively. If the editor agrees we would like to keep both, otherwise, we would redesign them.

  1. Sect. 3.1: who have identified seaweed? Please include this important information in the text of manuscript.

Response: This information is now available in the manuscript “Species identification was performed and double-checked by expert biologist both from Algamar and our laboratory”

  1. The authors' comments on my question 13 from the previous peer review round should be included in the manuscript as a limitation of this study.

Response: This reviewer’s comment was already included in the previous version of the manuscript (L 241-242) “In this sense, the solid:liquid (S:L) ratio may also be adjusted to maximize the use of the solvent, since the S:L suggested at the present work is hardly scalable at industrial sectors.” We consider this sentence already informs of the limitation regarding the S:L ratio.